# Recent Scientific Advancements towards a Vaccine against Group A *Streptococcus*

**DOI:** 10.3390/vaccines12030272

**Published:** 2024-03-05

**Authors:** Jingyi Fan, Istvan Toth, Rachel J. Stephenson

**Affiliations:** 1School of Chemistry and Molecular Biosciences, The University of Queensland, St. Lucia, QLD 4072, Australia; j.fan1@uq.net.au (J.F.); i.toth@uq.edu.au (I.T.); 2Institute for Molecular Bioscience, The University of Queensland, St. Lucia, QLD 4072, Australia; 3School of Pharmacy, The University of Queensland, Woolloongabba, QLD 4102, Australia

**Keywords:** Group A *Streptococcus*, peptide-based subunit vaccines, GAS vaccine candidates, preclinical studies, clinical studies

## Abstract

Group A *Streptococcus* (GAS), or *Streptococcus pyogenes*, is a gram-positive bacterium that extensively colonises within the human host. GAS is responsible for causing a range of human infections, such as pharyngitis, impetigo, scarlet fever, septicemia, and necrotising fasciitis. GAS pathogens have the potential to elicit fatal autoimmune sequelae diseases (including rheumatic fever and rheumatic heart diseases) due to recurrent GAS infections, leading to high morbidity and mortality of young children and the elderly worldwide. Antibiotic drugs are the primary method of controlling and treating the early stages of GAS infection; however, the recent identification of clinical GAS isolates with reduced sensitivity to penicillin-adjunctive antibiotics and increasing macrolide resistance is an increasing threat. Vaccination is credited as the most successful medical intervention against infectious diseases since it was discovered by Edward Jenner in 1796. Immunisation with an inactive/live-attenuated whole pathogen or selective pathogen-derived antigens induces a potent adaptive immunity and protection against infectious diseases. Although no GAS vaccines have been approved for the market following more than 100 years of GAS vaccine development, the understanding of GAS pathogenesis and transmission has significantly increased, providing detailed insight into the primary pathogenic proteins, and enhancing GAS vaccine design. This review highlights recent advances in GAS vaccine development, providing detailed data from preclinical and clinical studies across the globe for potential GAS vaccine candidates. Furthermore, the challenges and future perspectives on the development of GAS vaccines are also described.

## 1. Introduction

Group A *Streptococcus* (GAS) or *Streptococcus pyogenes* is a typical and virulent pathogen responsible for the induction of a wide range of mild, superficial infections to life-threatening diseases, including rheumatic heart diseases and acute rheumatic fever, leading to high global morbidity in humans [1]. Most of the GAS-associated mortality occurs in low-income countries and populations [2]. A host-adapted GAS pathogen adheres to the primary infection sites, including the nasopharyngeal mucosa of the throat and skin, from where it achieves colonisation and proliferation, destroying innate and adaptive immunity, and subsequently disseminating to a new host [3]. The production of various toxins or M proteins by GAS pathogens is related to GAS evading the host immune response, which can cause consistent dissemination of GAS infections with the potential induction of severe post-sequalae diseases [4]. GAS employs antiphagocytic virulence factors, including the M protein and the hyaluronic acid capsule, which effectively avoids the first line of defense from phagocytic cells [4].

In addition, *S. pyogenes* can survive in a variety of host environmental conditions to maintain its metabolic fitness [5]. Since the early 1980s, the epidemiological distribution of GAS infection has undergone significant changes in the world, particularly around invasive GAS infections due to specific M1T1 clones, dominantly occurring in most developed countries [6,7]. Populations at high risk of GAS infections include school-aged children and the elderly over 65 years of age, in addition to aboriginal people in the Pacific regions [8]. Additionally, in the past decade, changes in the bacterial genome and virulence of *S. pyogenes* made populations more susceptible to GAS-related diseases, leading to a sharp increase in several invasive GAS infections, such as scarlet fever and impetigo, attracting the attention of public health authorities worldwide [9,10]. Consequently, on the back of the COVID-19 pandemic, a sudden resurgence of severe invasive GAS infections in high-income countries is approximately 4-fold higher post-COVID-19 than that of pre-COVID-19 levels [11,12,13].

In the absence of a commercial GAS vaccine worldwide, the main medical intervention for controlling and treating GAS infections is the use of antibiotics [3]. Although antibiotics effectively control and treat non-invasive and superficial GAS infections, severe post-infection complications (e.g., rheumatic heart diseases and acute rheumatic fever) only use antibiotic prophylaxis to prevent post-sequelae GAS diseases or reduce the long-term severity of heart disease recurrence rather than treating these fatal diseases [14,15]. Severe post-infection complications, such as rheumatic heart disease and acute rheumatic fever are not treated by antibiotics [16]. Over the past few decades, GAS research has achieved reinvigorated progress in the fight against GAS infection, including the development of rapid diagnosis methods and effective GAS vaccines. This review highlights the global burden of GAS infection and the primary types of virulence factors of a GAS pathogen, with a focus on the most recent advances of GAS vaccines in preclinical and clinical development, and describes the potential challenges of developing a universal GAS vaccine.

## 2. Burden of GAS Infections

Group A *Streptococcus* (*S. pyogenes*) is a gram-positive bacterial pathogen that causes a range of illnesses, from non-invasive infections (e.g., impetigo, scarlet fever, and pneumonia) to severe invasive diseases (including bacteremia, necrotising fasciitis, and streptococcal toxic shock syndrome) and fatal post-GAS infections, such as acute rheumatic fever, rheumatic heart disease, and acute post-streptococcal glomerulonephritis [16,17,18,19,20]. The World Health Organisation indicated that at least 18.1 million individuals suffered from severe GAS diseases worldwide, with approximately 1.78 million new cases identified annually and more than 500,000 deaths occurring each year [21].

Rheumatic heart disease is a commonly acquired heart disease caused by an untreated or recurring GAS infection and is associated with high risks of morbidity, disability, and mortality [22,23]. A recent global burden of disease study estimated that approximately 33 million prevalent cases of rheumatic heart disease, leading to more than 9 million disability-adjusted life years lost and approximately 275,000 deaths each year [23]. The distribution of GAS infection significantly differed due to age and geographical area [24]. A high prevalence of rheumatic heart disease exists amongst young children and people over 65 years of age in low- and middle-income regions, including South Asia, Sub-Saharan Africa, and the Pacific islands [23]. In Australia, the epidemiology of acute rheumatic fever or rheumatic heart disease mainly affected Aboriginal and/or Torres Strait Islander and Pacific Islander children aged from 5 to 14 years of age living in regional and remote areas of central and northern Australia, with an estimated incidence of 250 to 350 diagnosed cases per 100,000 people [25,26]. Notably, in 2023, the Australian Institute of Health and Welfare reported there were approximately 9900 people living with the diagnosis of acute rheumatic fever and/or rheumatic heart disease in New South Wales, Queensland, Western Australia, South Australia, and the Northern Territory [27].

Since COVID-19, there has been an unusual rise in pediatric invasive GAS cases in high-income countries [28]. Starting in October 2022, several states across the United States, including Colorado, Minnesota, and Texas, found sharp increases in young children aged between 5 and 7 years of age suffering from invasive GAS infections, with an estimated 2500 to 9900 invasive GAS cases and 1100 to 2500 deaths annually [28]. Furthermore, the United Kingdom was credited as the worst-affected country for invasive GAS infections, with a sudden rise compared to other high-income countries [29]. As of December 2022, over 6600 cases of scarlet fever mainly affected children younger than 15 years of age in a period of just 12 weeks, along with approximately 650 more cases of invasive GAS disease presentations with 60 deaths, targeting mainly children aged between 10 and 14 years of age and the elderly aged over 75 years [9,29,30]. Simultaneously, multiple European countries, including France, Sweden, Ireland, Denmark, and Spain, reported a significantly increased trend of invasive GAS cases compared to the pre-COVID-19 pandemic period [9,29]. Thus, invasive GAS infections and post-GAS infections are threatening public health globally, particularly for school-aged children and the elderly over 65 years of age.

## 3. Virulence Factors of Group A *Streptococcus*

Serious and fatal human GAS infections are induced by the group A streptococci, which are complex and multifactorial, adapting to the various niches that are encountered in the human host, effectively avoiding systemic host defense [18,31]. Group A streptococci are capsulated by an outer hyaluronic acid capsule, while the group A carbohydrate antigen and the specific M protein are attached to the bacterial cell wall and membrane (Figure 1) [18].

Several key virulence factors, including pathogenic M protein, a hyaluronic acid, pyrogenic toxin, and GAS-related enzymes, determine the colonisation of GAS pathogens on epithelial tissues and elicit the progression of invasive or post-streptococcal infection sequelae [18,32].

### 3.1. Surface of Pathogenic M Protein

The M protein, a dominant virulence factor of GAS infection with more than 80 distinct serotypes identified, inhibits phagocytic functions in human tissues and fluids [33]. The GAS M protein was discovered and illustrated by Rebecca Lancefield in 1927 where it was identified as a primary source of strain-specific and long-term immunity to infecting serotypes [33,34]. The M protein is a typical alpha-helical coiled-coil dimer, composed of C- and N-terminal regions, hypervariable A, variable B, conserved C and D repeat regions, and an anchor region where its amino-terminal region extends from the surface of the streptococcal wall, while the carboxy-terminal region is within the membrane (Figure 1) [18,31].

The N-terminal region of the GAS M protein is furthest from the cell and contains long segments of negatively charged amino acids. Heterogenicity in amino acids present in the N-terminal region of the M protein results in antigenic diversity, leading to the formation of GAS serotypes, which are significantly different due to different regions [18,33]. Regions closer to the cell membrane are less diverse but also significantly variable and antigenic [33]. M proteins from the GAS pathogen in regions have a non-helical N-terminal region, variable central region, and highly conserved regions [17] The conservation of sequences increases are related to the D repeat region [17]. Additionally, a proline-glycine (Pro-Gly) rich region was intercalated into the bacteria cell wall, and a hydrophobic region acts as a membrane anchor [33,35]. GAS pathogens show significant intraspecies phenotypic variation, which are divided into type-specific serotypes in virulence and disease manifestation [36]. GAS M protein serotypes are affected by genetic changes over time due to alterations in virulence, geography, and antibiotic resistance factors. However, specific GAS diseases caused by M serotypes coded by emm gene types have expressed no significant changes over the past few decades [33] (Table 1).

Of the serologically classified M stereotypes, the most dominant epidemiology of GAS infectious diseases is related to the M1 stereotype [45]. Nine different emm types (including M1, 3, 5, 6, 14, 18, 19, 24, and 28) were detected among acute rheumatic fever case studies, of which the M18 GAS strain has been listed as the present main serotype of acute rheumatic fever outbreaks [43,46]. The existence of acute rheumatic fever induces severe inflammation of the heart, arthritis, and chorea, with the induction of rheumatic heart disease at a high prevalence worldwide [47]. M protein coats the surface of GAS bacteria. The essential virulence factor of GAS is defined as the M protein, which inhibits the phagocytotic functions of macrophages and maintains GAS pathogens’ survival in human tissues and fluids [33]. The whole structure of the M protein acts as the main antigen with the determination of specific immune responses [33]. However, specific regions of M protein also serve as peptide-based subunit antigens to induce adaptive immunity and eliminate autoimmune responses, allergic responses, and inflammations caused by the use of whole M proteins as an antigen [33,48]. Therefore, immune reaction to GAS M protein provides protection against GAS infection, which encourages the development of universal GAS vaccines directed against M protein. Antigens of peptide-based subunit vaccines are mostly selected from conservative regions of the M protein (e.g., C terminal region) instead of the N-terminal variable regions [33,49,50].

### 3.2. Hyaluronic Aicd as a Non-M Protein GAS Virulence Factor

The hyaluronic acid capsule is a polymer of hyaluronic acid composed of repeating units of glucuronic acid and N-acetylglucosamine, making the mucoid colony morphology smooth, watery, and of regular contour [18,51,52]. Three different genes including *hasA*, *hasB*, and *hasC* are involved in the production of the hyaluronic acid capsule which are completed by the same operon [18,53]. Synthesis of hyaluronate synthase and glucose dehydrogenase are encoded by *hasA* and *hasB*, respectively [18]. The formulation of glucose phyrophosphorylase is controlled by *hasC* [54]. However, only *hasA* and *hasB* are significantly required for the expression of the capsule in group A *streptococci* [18]. Different streptococcal isolates secrete an amount of hyaluronic acid capsules, which are related to the operon promotor [18]. The promoter was more active in a well-encapsulated strain while less active promoters were in a poorly encapsulated strain [18,55]. The highly mucoid hyaluronic acid capsule induces the effective and smooth adherence of GAS pathogens in the pharynx, allowing it to bind to the surface glycoprotein CD44 of human epithelial cells, enhancing the colonisation and proliferation of GAS pathogens, resulting in invasive GAS infections and post-infections of GAS sequelae [18,52,56,57]. In addition, human hyaluronic acids are structurally similar to the hyaluronic acid capsule of the GAS pathogen, which is found in extracellular matrices present in many human tissues (e.g., connective and epithelial tissues) [3]. The hyaluronic acid capsule, typical of GAS virulent factors, is anti-phagocytic providing potent resistance to phagocytosis by natural killer cells in evasion of the host immune system [58,59]. At the turn of the 19th century, Ward and Lyons et al. indicated that the resistance to phagocytosis of virulent strains is remarkably implicated in the presence of capsules on the organisms [52]. The physical barrier of the capsule expresses capabilities to prevent access of phagocytes to recognise and eradicate opsonic complement proteins on the bacterial surface [18,60]. Although the hyaluronic acid capsule is identical to the polysaccharide in humans and is considered a weak immunogen, anti-hyaluronate antibodies are detected after immunisation of animals. More recent work provides definitive evidence that the capsule is the predominant virulence factor of GAS due to resistance to phagocytosis [18].

### 3.3. Main Pyrogenic (Erythrogenic) Toxins as Antigens in GAS Vaccine Development

Non-M protein pyrogenic toxins play a crucial role in the colonisation, pathogenesis, and dissemination of GAS pathogens, including hemolysins and superantigens, of which a number have been utilised as vaccine antigens or as novel therapeutic methods [61].

β-haemolysis was initially discovered in 1895 by Ayer et al. [61]. The effective lysis of red blood cells by pyrogenic toxin β-haemolysis, and was expressed by the characteristic area of clearing surrounding bacterial colonies on the surface of the blood agar medium [61]. GAS isolates at the stationary phase express abilities to secrete the potent membrane-active hemolysin, named streptolysin S (SLS), which are a small and oxygen-stable exotoxin [62]. SLS exotoxin targets and destroys the function of multiple host cells and acts by the formation of a pore in the hose cell membranes [63]. In 2016, Higashi and colleagues used high-resolution live cell imaging in which SLS induced a dramatic osmotic change in red blood cells, disrupting the function of anion transporters to cause an influx of chloride anions (Cl^−^) and cell lysis [63,64]. Additionally, SLS conjugated with the host protease calpain to induce the destruction of intracellular junction proteins, producing paracellular invasion of the bacteria across epithelial barriers [63,65]. Accordingly, the pathogenicity of SLS contributes to inhibiting phagocyte function and damaging epithelial barriers [63]. Streptolysin O (SLO) is an oxygen-sensitive, well-characterised, and cholesterol-binding β-haemolysis exotoxin [66]. SLO, a pore-forming toxin, is divided along the thiol-activated family of cytolysins, additionally, SLO determines the pathogenesis of GAS expression during the exponential phase and early stationary growth phases [66,67]. SLO cytolysin is capable of destroying the integrity of cytoplasmic membranes in various eukaryotic cell types, including erythrocytes, leukocytes, macrophages, epithelial cells, and tissue culture cell lines [61,68]. Furthermore, the *slo* gene is transcribed with the *nga* gene to encode NAD-glycohydrolase, which is actively translocated into the cytosol of human epithelium by SLO to deplete energy stores, destroying host cells against invasive GAS pathogens [61]. Michos et al demonstrated that the expression of both NAD-glycohydrolase and SLO together significantly enhanced the lytic activity for erythrocytes by exotoxins but had no effect on the SLO-mediated portion of synthetic cholesterol-rich liposomes. [69]. *S. pyogenes* is one of a small number of bacterial species to produce mitogenic superantigen (SAgs) exotoxins, which are low molecular weight exotoxins (22 to 28 kDa) [62]. Superantigens function by the induction of excessive T cell activation. Superantigens simultaneously bind to both the variable region of the T cell antigen receptor (TCR) β-chain on T cell and the major histocompatibility complex (MHC) outside the antigen-binding cleft on antigen-presenting cells, with the release of T cell mediators and pro-inflammatory cytokines, causing various infections [62,70,71]. Since 1924, numerous streptococcal superantigens have been discovered, prophage-encoded three types of streptococcal pyrogenic exotoxin (Spe) superantigens, including SpeA, SpeC, and SSA have been literature as crucial virulent strains of inducing, toxic shock syndrome, scarlet fever, severe invasive disease and acute immunodeficiency disorders by GAS [3,61,72,73].

### 3.4. Enzymes Involved in GAS Pathogenesis

Extracellular enzymes are secreted by GAS pathogens to interact with the host immune system [74]. These streptococcal enzymes (e.g., streptokinase, Sep B, C5a peptidase, and other enzymes) are capable of modulating the activity of immune defense molecules, including immunoglobulins, complement factors, cytokines, and/or inflammatory medicators, leading to invasive GAS infections or post-infections sequels [74].

Streptokinase is a typical single-chain virulent enzyme composed of 414 amino acids to activate the host protein plasminogen, forming a complex with the inactive zymogen plasminogen [62,75]. Activator streptokinase-plasminogen complex binds to the surface of GAS bacteria, activating the contact system resulting in the release of bradykinin, triggering to proteolysis of host defense proteins, damaging widespread human tissues and promoting further dissemination of the bacteria [62,76,77,78].

Streptococcal pyrogenic exotoxin B (SpeB) is a highly conserved cysteine proteinase in virtually all GAS pathogens [62]. SpeB is a predominant extracellular and broad-spectrum protease, which is secreted as a 42-kDa zymogen and autocatalyzed into an active 28-kDa cysteine protease [79,80]. As a super-active protease with endopeptidase activity, SpeB contributes to the degradation of host extracellular matrix, immunoglobulins (IgG), complement components, GAS surface adhesins (including M protein and protein F1), multiple superantigens, leading to the escape of bacteria from immune clearance, invade the deeper tissue, and disseminate bacteria from the primary infection sites to host cells [79]. High-level production of SpeB protease is involved in the outbreak of acute post-streptococcal glomerulonephritis, scarlet fever, streptococcal toxic shock syndrome, and fatal post-GAS infections [79]. C5a peptidase (SCPA) from *S. pyogenes* is a type of serine endopeptidase that is anchored on the surface of the bacteria and specifically cleaves the chemotactic complement factor C5a, triggering to inhibit recruitment of phagocytic cells to the infectious site and effectively activate neutrophils secreted from phagocytic bacteria, disrupting defense by systemic immunity [74,81,82,83]. Moreover, intranasal administration with recombinant SCPA in mice stimulated potent humoral and mucosal immunity and achieved the reduction of nasopharyngeal bacterial colonisation [84]. Although streptokinase, SpeB, and C5a peptidase are predominant GAS enzymes from GAS pathogens to negatively affect the induction of systemic immunity, Table 2 listed other enzymes that inhibit the host immune response, disseminating bacteria from colonising sites to host cells.

A number of extracellular enzymes are produced by *S. pyogenes* [74]. Several of these enzymes (such as SpeB, C5a peptidase, Endos, and Ides) host-parasites interact with the host immune system, circumventing host defense mechanisms to successfully colonise and disseminate within a host [74]. Additionally, immunoglobulins IgG serves as a substrate for some extracellular enzymes (including SpeB, IdeS, and Endos), eliminating the activity of IgG through binding with enzymes, and conferring the loss of adaptive immune response [74]. Therefore, the modulation of extracellular enzymes is considered the pathway to terminate GAS dissemination [74].

## 4. Recent Advances of GAS Vaccine Development

GAS is a ubiquitous pathogen responsible for a tremendous global disease burden. The majority of therapeutic methods for the treatment of non-invasive GAS infections include the use of antibiotics (e.g., penicillin), however, antibiotic resistance confers to poor therapeutic results in first-in-line adjunctive, penicillin alternate treatments, and subclinical β-lactam treatment, leading to recurrent GAS infections with treatment failure. Therefore, the difficulty of antibiotic-treating post-infection sequelae GAS infections necessitates elimination through the development of an effective GAS vaccine [3,91]. Furthermore, geographic distribution and various GAS *emm* genes cause a wide spectrum of GAS isolates, capable of reducing the susceptibility of penicillin and increasing macrolide resistance, threatening both frontline and penicillin-adjunctive antibiotic treatment [3]. Despite GAS vaccine research since the early 20th century, no commercial GAS vaccine has been developed [17]. Autoimmune responses, allergenicity, and inflammations failed to deliver safe and efficient products against GAS infection since the use of live-attenuated or inactivated GAS bacteria as vaccine antigens [92]. Hence, subunit vaccines, a recent development in the vaccine space, are composed of fragments of pathogens (e.g., protein, glycoprotein, peptide, or carbohydrate) from a pathogen, as an antigen, inducing adaptive and protective immunity against diseases [93]. Peptide-based subunit vaccines use a minimal microbial antigenic peptide sequence derived from a pathogenic protein to stimulate adaptive immunity against a pathogen, eliminating unwanted inflammation, and allergic and/or autoimmune responses [17]. Thus, the design of peptide-based subunit vaccines involves two key steps, including the identification and selection of epitopes, and the enhancement of poor immunogenicity.

M proteins, the predominant virulence factor of GAS infections, are the most common pathogenic antigens of GAS vaccine candidates in both preclinical and clinical phases. Other vaccine candidates that are in preclinical trials contain numerous non-M protein virulence factors such as various enzymes, erythrogenic toxins, and capsule acid. Moreover, numerous virulence factors (e.g., various enzymes, erythrogenic toxins, and capsule acid) are classified as non-M protein antigens, triggering the induction of potent immunity against GAS in preclinical trials [17,94]. Here, we reviewed preclinical trials of M-protein and non-M protein-based GAS vaccines in the last decade (Table 3) highlighting recent advances in clinical trials of GAS vaccine candidates (Table 4).

**Table 3 vaccines-12-00272-t003:** Recent decade-update on preclinical trials of GAS vaccine candidates.

Vaccine Candidate	Target Antigen	Adjuvant	Results	Reference
**Non-M protein vaccine candidates**				
➢ **Single non-M protein subunit vaccine**				
Group A carbohydrate (GAC)	Trirhamnosyl-lipopeptide	-	Enhancement of immunogenic activity in mice and 75–97% opsonic activity against four different GAS clinical isolates.	[95]
	Lancefield antigen (GAC lacking GlcNAc side chain)	-	Promotion of opsonophagocytic killing of multiple GAS serotypes and protection against systemic GAS challenge after immunisation	[96]
A transpeptidase Sortase A	Sortase A	CTB	Increase of CD4^+^ IL-17A^+^ cells in nasal-associated lymphoid tissue and the induction of high levels of Sortase A -specific antibodies	[97]
C5a peptidase	Inactive enzyme-ΔSCPA^H193A^	Alum	Stimulation of high titres of antigen-specific IgG1 antibodies and potent T cell-mediated immunities	[98]
Streptococcal pyrogenic exotoxin A/B	SpeA mutants	Alum	Production of SpeA-specific IgG and low risk of suffering from STSS	[99]
	SpeAB fusion protein	Alum	Stimulation of both SpeA and SpeB-specific antibodies	[100]
Streptococcal pili	*L. lactis* PilM18	-	Induction of both specific IgG antibodies and significant mucosal immune response against GAS strains T18 and T28, respectively	[101]
	*L. lactis* PilM28			
Calprotectin	Adhesin competence repressor (AdcR),	Alum	Calprotectin imposes zinc limitation on GAS pathogens during infection and reduces of growth of GAS pathogens.	[102]
➢ **Multicomponent non-M protein subunit vaccines**				
C5a peptidase-conjugates	Sortase A/SCPA	CTB	Induction of mucosal Th17 and robust SCPA antibody responses through rapid infiltration and activation of neutrophils.	[103]
	GAC oligosaccharide-ΔSCPA^H193A^	CTB	Secretion of CD4^+T^ cells and helper T cells to differentiate into antigen-specific Th1 and Th2 cells.	[104]
	SCPA-Fn domain	IFA	Induction of high levels of IgG1 and IgG2 titres with cross-reactivity to SCPA.	[105]
GAC-arginine deiminase (ADI)	ΔGAC lacking GlcNAc side chain-conjugated to ADI)	Alum	High-titre antigen-specific antibody responses with bactericidal activity.	[106]
Combo#5	Combo5 (ADI, SCPA, SLO, SpyCEP, TF)	Alum Advax2 Advax4 SWE LQ LMQ SMQ	Alum reduced bacterial dissemination to the spleen and blood. Different adjuvant selection in raising balanced Th1/Th2-type response and inducing long-term protective immune responses against GAS invasive infection.	[106,107]
Spy7	SCPA, OppA, PulA, Spy1228 ^a^, Spy1037 ^a^, Spy0843 ^a^, SpyAD	CFA/IFA	Low dissemination of GAS M1 via the production of anti-streptococcal antibodies and the modulation of IL-8 cytokine release.	[108]
5CP	SrtA, SCPA, SpyAD, CEP-5, SLO	CpG	Induction of mucosal and systemic immunity against GAS across serotypes with no autoimmune response.	[109]
VAX-A1	GAC^PR^-, SpyAD, SLO, SCPA	Adju-phos^®^	Promotion of opsonophagocytic killing by human neutrophils and secretion of IgG titres.	[110]
TeeVax	TeeVax 1–3 combinations	IFA /Alum	Production of a robust antibody response in rabbits with cross-reactive 21 T-antigens, is expected to provide over 95% vaccine coverage.	[111]
**M protein vaccine candidates**				
➢ **N-terminal peptide-based subunit vaccines**				
Mx10	A 10-valent vaccine	-	Induction of high levels of serum and bronchoalveolar lavages IgG titres and low colonisation of bacteria strains on oropharyngeal.	[112]
➢ **Conserved C-repeat region peptide-based subunit vaccines**				
Synthetic P*17 epitope	P*17	Alum	Induction of more antibody titres and potent p145-specific immune response and significantly better protection from GAS infections on the surface of the skin compared to the p145 peptide itself.	[113]
	P*17-DT+K4S2-DT	CAF^®^01	Induction of high levels of IgG titres and mucosal immunity, with the significantly decreasing colonisation of GAS bacteria in the upper respiratory tract and skin.	[114]
	P*17-CRM+K4S2-CRM	Alum	Induction of p145-specific IgG antibodies with long-lasting proactive capacity and no evidence of vaccine-related toxicological and safe issues in rats.	[115]
Synthetic J8 epitope	J8-based lipopeptide (J8-PADRE conjugated with polyglutamic acid)	Capsulated by trimethyl chitosan	Induction of robust systemic and mucosal antibody titres and J8-specific antibodies were also opsonic against clinically isolated GAS strains.	[116]
	J8-PADRE conjugated with poly-hydrophobic amino acid (pHAA) self-adjuvanting peptide-based vaccine candidate	-	PHAA, a self-adjuvanting peptide, induced strong J8-specific IgG response and mucosal immunity, with the highest opsonisation of GAS clinical isolates.	[117]
	J8-PADRE conjugated with linear poly-hydrophobic amino acid (pHAA) self-adjuvanting peptide-based vaccine candidate	-	Linear conjugate bearing J8-PADRE and 15 polyleucines induced robust IgG antibody response and long-term lasting protective immune response compared to commercial adjuvants.	[118]
	Physically mix J8-PADRE with cyclic peptide and lipid peptide	-	Induction of a potent and balanced J8-specific immune response by the physical addition of cyclic lipopeptide.	[119]
Synthetic J8 peptide sequence	Synthetic J8-DT peptide/HD-MAP (J8-DT delivered by a high-density microarray patch (HD-MAP))	HD-MAP	Induction significantly higher anti-J8 IgG titres compared to J8-DT/Alum, and reduced skin and blood bacterial burden.	[120]
	J8-DT+ CXC chemokine protease (J8-DT combined with inactive CXC chemokine protease)	Alum	Highly effective inhibition of chemokine degradation and production of opsonic antibody to kill the bacteria via a combination of J8-DT with inactive CXC chemokine protease	[121]
	J8-DT+S2 (J8-DT combined conjugates of minimal epitope (S2) from rSpyCEP with DT	Alum	Induction of anti-S2 and anti-J8 antibodies against hypervirulent GAS strains.	[122]
	J8-CRM+K4S2-CRM_197_	Alum	Reduction in streptococcal load in blood from systemic infection with systemic toxicity.	[115]
Synthetic J14 epitope	P25-J14-C16C16 lipopeptides	-	Production of high levels of J14-specific antibody with potent humoral and mucosal immunity.	[123]
	Lipid core peptide including P25-J14	bis-(3′,5′)-cyclic dimeric adenosine monophosphate (c-di-AMP) and BPPCysMPEG,	Induction of potent mucosal and humoral immune response by the use of two different adjuvants.	[124]
	J14 conjugated to FSL-1	-	Induction of strong J14-specific immune response and coffered protection against GAS strain 5448 compared unconjugated J14/FSL-1 with alum.	[125]

a: Stop at the residue 1228, 1037, and 0843. P*17 based on the highly conserved C3-repeat region of M protein.

### 4.1. Non-M Protein-Based Vaccine Candidates in Preclinical Trials

Non-M protein virulence factors have contributed to numerous preclinical trials in different animal models against GAS challenges, which have been summarised for non-M protein-based candidates in Table 3. Group A carbohydrate is located on the cell wall of GAS bacteria and used as the primary single non-M protein antigen in GAS vaccine development. Here, Khatun et al. created a trirhamnosyl lipopeptide vaccine candidate which was composed of a single carbohydrate repeating unit of GAC, universal T-helper peptide epitope T-helper pan-DR helper T lymphocyte (PADRE), and lipid core peptide adjuvant, leading to the induction of strong antibody tires and 75–97% opsonic activity against four different GAS clinical strains [95]. In addition to this, a classical Lancefield antigen containing the N-acetylglucosamine side chain of GAC has been used in a GAS vaccine where the Lancefield antigen alone was intraperitoneally immunised into mice, with the induction of innate immunity and control of serotype M1 GAS dissemination [96]. Over the last decade, some non-M protein virulence factors selected from the GAS bacteria (e.g., inactive enzyme ΔSCPA^H193A^, SpeA derivatives calprotectin) were single antigens co-administrated with alum adjuvant following subcutaneous immunisation in mice, leading to high antibody titers and a reduction in the growth of GAS pathogens [98,99,100,102].

More recent achievements of non-M protein vaccine candidates against GAS infections focus on the use of multicomponent non-M protein antigens, conferring the promise of a broad spectrum of GAS strains and enhancing vaccine efficacy. Here, recent non-M protein multicomponent vaccines with indicated efficacy in animal models following Table 3.

The Combo#5 vaccine is composed of a combination of SLO, SpyCEP, SCPA, and two highly conserved antigens, arginine deiminase and trigger factor, selected from anchorless streptococcal bacteria [106,107]. Rivera-Hernandez and co-workers identified that this Combo#5 vaccine was adjuvanted with alum, and after subcutaneous immunisation in mice, the mouse serum detected high levels of antigen-specific antibody with bactericidal activity. A further preclinical study of evaluation of the immunological capability of Combo#5 formulated with a panel of 7 different adjuvants (including alum, Advax-2, Advax-4, squalene-in-water-emulsion, neutral liposomes, neutral liposomes containing saponin QS21, and SMQ) was conducted by Rivera-Hernandez el al. in 2020 [107]. This study indicated that the addition of SMQ adjuvant composed of cholesterol, Monophosphoryl lipid A (MPL), and saponin QS21 provided significant protective efficacy, generating a strong antigen-specific antibody immune response compared to other vaccines formulated with 6 adjuvants, with the induction of Th1 immune response [106,107].

The Spy7 vaccine composed of 7 different selected antigens, including C5a peptidase, oligopeptide-binding protein, pullulanase, nucleoside-binding protein, hypothetical membrane-associated protein Spy0762, cell surface protein Spy0651 and SpyAD, was adjuvanted with Freund’s adjuvant. Intramuscular administration of the Spy7 vaccine in mice produced specific anti-streptococcal antibodies and limited systemic dissemination of M1 and M3 GAS isolates [108]. A five-component GAS vaccine named 5CP included sortase A (SrtA), SCPA, SpyAD, SpyCEP, and SLO. Bi et al., showed intranasal 5CP co-administration with an adjuvant CpG in mice leading to Th17 and antibody response locally and systemically in murine nasal-associated lymphoid tissue [109]. A multivalent non-M protein vaccine, VAX-A1 composed of SpyAD-GAC^PR^, SCPA, and SLO antigens stimulated systemic immunity with the goal of broad protection of GAS strains without autoreactivity following the intravenous administration route in mice [110]. A type of recombinant multivalent protein vaccine is TeeVax selected antigenic protein T-antigen from the GAS pilus. Subcutaneous vaccination with TeeVax1-3 induced a robust specific T-antigen antibody immune response in rabbits with a broad coverage of GAS strains and significant cross-activity with a full panel of T-antigens [111].

### 4.2. M Protein-Based Vaccine Candidates in Preclinical Trials

Over the past decade, various peptide-based antigens selected from the pathogenic M protein achieved considerable progress in clinical trials in peptide-based subunit vaccines, with a summary of recent preclinical trials of peptide-based subunit vaccine candidates in Table 3. A multivalent vaccine, containing 10 valents, from the hypervariable N-terminal regions of M protein by Wozniak et al. [112]. Here, Wozniak and co-workers developed an intranasal live bacterial vaccine composed of 10 strains of *Lactococcus lactis* expressing one fragment of M protein, with the induction of high levels of serum and bronchoalveolar lavages IgG titres and low colonisation of bacteria strains on oropharyngeal [112]. Protective immune responses against infections are associated with antibody secretion directed against the N-terminal or C-terminal region of the M protein. The hypervariability of the N-terminal region poses a challenge in the development of a vaccine against all serotypes of GAS. However, a vaccine strategy developed from the conserved C terminal region of M protein is universal against all GAS serotypes, overcoming restrictions of GAS vaccines designed from N termini [49,50] The initial discovery of the B cell epitope from the GAS M protein was a small peptide, p145 (LRRDLDASREAKKQVEKALE), containing 20 amino acids derived from the conserved C3 repeat region of the M protein. However, it induced autoimmune responses [17,126,127]. Therefore, the p145 epitope was modified into two smaller B cell epitopes called P*17 and J8 which induce a humoral immune response against GAS infection and eliminate autoimmune response. Additionally, the 14J epitope is a close analog of J8 against GAS global challenges, expressing no cross-reactivity with human tissues.

Preclinical research by Nordstrom et al. assessed levels of IgG antibody in the serum of mice following vaccination with P*17 peptide adjuvanted with alum. This research identified that the P*17 peptide induced strong IgG titres and produced significant opsonic reactivity of two GAS clinical strains with the elimination autoimmune response compared to the p145 peptide alone [113]. Reynolds et al. developed a candidate vaccine comprised of synthetic M protein P*17 epitope and non-M protein epitope (K4S2) where each epitope was conjugated to the carrier protein CRM_197_ and adjuvanted with alum to formulate the final vaccines P*17-CRM+K4S2-CRM/alum. Mice intramuscularly immunised with three doses of P*17-CRM+K4S2-CRM/alum resulted in the production of a potent and sustained immunological response [115]. Furthermore, Ozberk et al. also used the P*17 epitope and non-M protein epitope (K4S2) to formulate a combination vaccine, P*17/K4S2, which were individually conjugated with mutant diphtheria toxin (DT) and administrated with approved adjuvant, CAF^®^01, to produce the final vaccine, P*17-DT+K4S2-DT/CAF^®^01 [114]. The immunogenicity and efficacy of this P*17-DT+K4S2-DT/CAF^®^01 vaccine were evaluated in mice following intranasal immunisation, and shown to elicit high levels of IgG titres and mucosal immunity, with a significantly decreasing colonisation of GAS bacteria in the upper respiratory tract and skin [114].

Here, the J8 epitope is another primary antigenic peptide that was conjugated with the universal T helper PADRE epitope, which has been applied in several preclinical studies for the development of self-adjuvanting peptide-based vaccines [48,92]. Nevagi et al. conjugated the J8-PADRE antigen with poly-glutamic acid and formulated these into nanoparticles with chitosan by ionic interactions, and mice intranasally immunised with this nano-vaccine gave potent systemic and mucosal immunity as detected in mice serum [116]. Furthermore, Skwarczyski and co-workers coupled J8-PADRE with poly-hydrophobic amino acids (PHAA) comprised of 15 leucine residues severing as a self-adjuvanting vaccine [117]. Following subcutaneous immunisation into mice, this PHAA self-adjuvanted vaccine induced a robust J8-specific immune response and produced a significant opsonised reaction with GAS clinic strains [117]. A further study evaluated the efficacy of linear structure poly-hydrophobic amino acids (PHAA) conjugated with J8-PADRE in the murine model, resulting in the stimulation of the highest levels of IgG antibody titres compared to commercial adjuvants [118]. Additionally, a physical mixture of J8-PADRE epitope with lipid peptide and cyclic decapeptide, as a vaccine candidate, was subcutaneously administrated into mice. The preclinical trial of this physical mixture induced a significantly higher J8-specific immune response compared to conjugation groups [119]. The most recent study of J8 was the evaluation of the J8-DT (J8 epitope conjugated with diphtheria toxin) vaccine, delivered by a high-density microarray patch (HD-MAP). Mills et al. observed that vaccination with J8-DT coated by HD-MAP induced similar levels of IgG antibody responses to vaccination J8-DT adjuvanted with alum [120]. Additionally, Pandey et al. combined synthetic peptide J8-DT with an inactive form of the streptococcal CXC protease, adding alum adjuvant and formulating a combination GAS vaccine, which was intraperitoneally administrated into mice [121]. The high increase of chemokine secretion with the strong protective immune response against pyoderma and bacteremia was expressed in the preclinical study [121]. In the further study, J8-DT combined conjugates of minimal epitope (S2) from SpyCEP [122]. Following subcutaneous administration in a preclinical trial mice induced anti-S2 and anti-J8 antibodies against hypervirulent GAS strains with long-term protective immunity [122].

A J14 epitope was used to develop self-adjuvanting peptide-based subunit vaccines [128]. Marasini and co-workers developed a self-adjuvanting lipopeptide-based (LCP-1) vaccine candidate containing a J14 epitope, P25 T-helper epitope, and lipid moieties, which was formulated into nanoparticle size [123]. The immunogenicity of this self-adjuvanting lipopeptide-based was evaluated in mice and high levels of J14-specific salivary mucosal and systemic immune response were detected after intranasal immunisation [123]. Further improvement of the immunogenicity of LCP-1 was achieved using different types of adjuvants, such as C-di-AMP and BPPCysMPEG, respectively [124]. The adjuvanticity C-di-AMP co-administrated with the J14 epitope produced a potent systematic mucosal immune response, allowing for a reduction of antigen dose compared to the BPPCysMPEG adjuvant [124]. Furthermore, Xu and co-workers conjugated TLR-2 agonist with J14 antigen to formulate fibroblast-stimulating lipopeptide-1 (FSL-1), with intramuscular administration into mice. The result of anti-J14 specific IgG antibodies was higher compared to a mixture of the J14 antigen alone with alum adjuvant and conferred a protective immune response against GAS strains [125].

### 4.3. GAS Vaccine Candidates in Clinical Trials

All vaccine candidates entered into the clinical phase target the GAS M protein where the main antigens are from the hypervariable N-terminal fragments to various M serotypes or conserved epitopes derived from the C-terminal region of the M protein (Table 4).

**Table 4 vaccines-12-00272-t004:** Clinical phases of GAS vaccine candidates.

Vaccine Candidate	Phase	Year	Adjuvant	Results	Reference
Hexavalent amino-terminal M protein polypeptide	1	2004	Alum hydroxide	Recombinant hexavalent M protein.	[129]
StreptAvax	1	2005	Alum hydroxide	4 recombinant proteins adsorbed to alum hydroxide which is composed of N-terminal peptides from streptococcal protective antigen and M proteins of 26 common pharyngitis, invasive, and/or rheumatogenic serotypes.	[130]
	2	2006	Alum hydroxide		
MJ8VAX	1	2018	Alum hydroxide	J8 epitope from the C-terminal region of M proteins conjugated with K4S2 from non-M protein SpyCEP.	[131]
P*17/S2 combivax	1a	2019	Alum hydroxide	P*17 peptide from C-terminus conjugated with a 20-mer K4S2 from SpyCEP.	[132]
StreptAnova	1	2020	Alum hydroxide	Protein-based subunits containing the N-terminal regions of M proteins from 30 GAS stereotypes.	[133,134]

The most advanced N-terminal peptide-based vaccine, StreptAnova was assessed in phase I clinical trials in 23 healthy volunteers in 2020 [133,134]. Outcomes for this phase 1 clinical trial indicated that intramuscular administration of StreptAnova adjuvanted with alum was well tolerated and no clinical evidence of autoimmune response and no laboratory evidence of tissue cross-reactive antibodies [133,134]. The immunogenicity of StreptAnova in healthy volunteers secreted IgG antibodies and induced strong opsonophagocytic killing activity in selected GAS strains [133,134]. Furthermore, approval for a phase 1a clinical trial of P*17/S2 combivax and J8/S2 combivax intramuscularly immunised into healthy volunteers was commenced, however, to date, no further information is available from this trial [132]. A phase 1 clinical trial of MJ8VAX vaccination (containing the J8 peptide antigen conjugated to DT and formulated with alum) was conducted in 10 healthy volunteers. This clinical trial showed that the two-dose administration of MJ8VAX was safe and tolerated, with no local reactogenicity recorded in participants. MJ8VAX vaccine induced specific-J8 antibodies and increased the secretion of anti-DT serum antibodies [131]. Moreover, Phase 1 and 2 clinical trials of StreptAvax were conducted in healthy adults aged between 18 and 50, with significant safety and efficacy, however, no updated data on this vaccine since 2005 [130].

## 5. Conclusions

Group A *Streptococcus* infections have a high prevalence worldwide. Patients from low-income countries contribute to the mortality of GAS infections due to poor diagnosis and treatments. However, since 2021 some high-income countries have also experienced sudden outbreaks with high mortality of school-aged children. The pathogenic M protein is considered a crucial antigen of GAS vaccines in addition to several non-M protein fragments. Advances in GAS vaccine development in preclinical and clinical settings have shown in the last decade that M protein-based GAS vaccines have a greater potential than non-M protein-based vaccines. Peptide-based subunit vaccine uses minimal peptides as an antigen (including J8, J14, and P145), which are selected from pathogenic M protein, inducing systemic and long-term immunity, and eradicating autoimmune response, inflammations, or allergic immunity compared to traditional vaccines. Although the low immunogenicity of peptide-based subunit vaccines necessitates potent immunostimulators to induce a potent adaptive immune response against GAS infections, peptide-based subunit vaccines showed the safety and efficacy of vaccine candidates in preclinical and early clinical studies. Overall, numerous preclinical trials of potential vaccine candidates achieved substantial progress in different animal models. However, the clinical development of GAS vaccines has hallmark restrictions compared to pre-clinical trials due to several inevitable reasons, including the absence of defined human immunity correlates of protection, insufficient animal models, variety of GAS strains, and perceptions of the risk of autoimmune complications triggered by the vaccine [135]. Therefore, the development of GAS vaccines necessitates global efforts via supporting preclinical and clinical trials to identify a universal and effective GAS candidate to prevent GAS infections.

## Figures and Tables

**Figure 1 vaccines-12-00272-f001:**
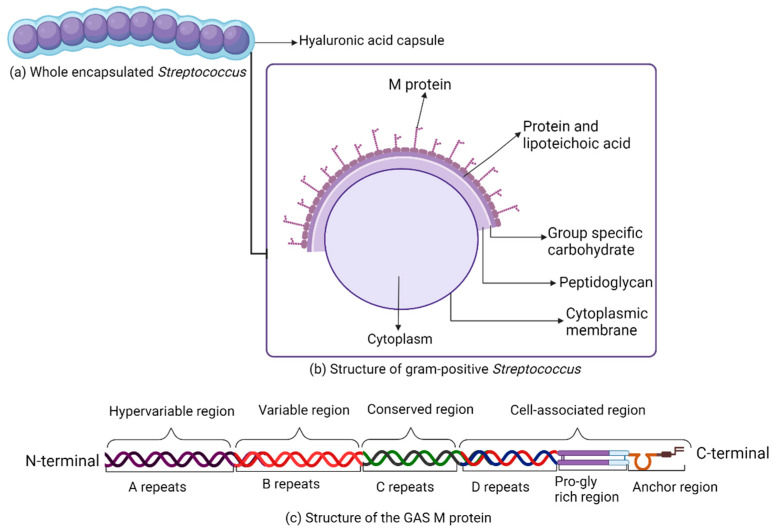
Gram-positive *Streptococcus pyogenes* schematic. (**a**) Whole encapsulated *Streptococcus*; (**b**) structure of *Streptococcus pyogenes*; and (**c**) structure of the GAS M protein.

**Table 1 vaccines-12-00272-t001:** Main M-protein serotypes in predominant GAS infections and post-sequalae diseases.

Clinical Diseases	GAS Serotypes of M Proteins	Ref.
Pharyngitis	M1, M3, M5, M6, M12, M14, M17, M19, M24	[18]
Scarlet fever	M4, M28, M1, M3	[37]
Impetigo	M33, M41, M42, M52, M53, M70	[18,33,38]
Meningitis	M1, M12	[18,33,39]
Streptococcal toxic shock syndrome	M1, M3	[40]
Acute post-streptococcal glomerulonephritis	M1, M2, M12, M55, M49, M60, M73	[41]
Acute rheumatic fever	M1, M3, M5, M6, M14, M18, M19, M24, and M29	[42,43]
Rheumatic heart disease	Unknown	[3,44]

**Table 2 vaccines-12-00272-t002:** Enzymes associated with the induction of GAS infections.

Types of Enzymes	Enzyme Name	Activities	Ref.
Serine protease	*S. pyogenes* cell envelope proteinase (SpyCEP)	Cleavage of the chemokine interleukin (IL)-8	[74,85]
Glycoprotein endonuclease	Deoxyribonucleases (DNases)	Alteration neutrophil migration	[74,86]
β-N-acetylglucosaminidase	Endos	Hydrolysation of the glycan on IgG	[87]
Cysteine protease	Ides/Mac-1, Mac-2	Cleavage of IgG and inhibition of opsonophagocytosis	[88]
Arginine deiminase	Streptococcal acid glycoprotein (SAGP)	Inhibition of T-cell proliferation	[89]
Superoxide dismutase	Manganese-dependent superoxide dismutase (SodA)	Protection of GAS bacteria from exogenously produced reactive oxygen species	[90]
Extracellular protease	Streptokinase	Release of bradykinin and targeting to proteolysis of host defense proteins	[77]
Cysteine proteinase	Streptococcal pyrogenic exotoxin B (SpeB)	Degradation of host extracellular matrix, immunoglobulins (IgG), complement components, GAS surface adhesins	[79]
Serine endopeptidases	C5a peptidase (SCPA)	Inhibition of recruitment of phagocytic cells to the infectious site and activation of neutrophils secreted from phagocytic bacteria	[82,83]

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
