# Peer review of "Recent Scientific Advancements towards a Vaccine against Group A Streptococcus"

_vaccines, 2024, doi:10.3390/vaccines12030272_

Round 1

Reviewer 1 Report

Comments and Suggestions for Authors

Dear authors, I have read your manuscript Recent scientific advancements towards a vaccine against group A Streptococcus, and these are my comments and suggestions:

Lines 12-13: Change to "Group A Streptococcus (GAS), or Streptococcus pyogenes, is a gram-positive bacterium that extensively colonises the human host. It is responsible for causing a range of human infections such as pharyngitis, impetigo, scarlet fever, septicaemia, and necrotising fasciitis."

Line 23: Change to "have been approved".

Line 31: Change "Mainly infected on children and the elderly" to "mainly infects children and the elderly".

Lines 40-41: Why would GAS disseminate to a new host after destroying innate and adaptive immunity in the old host? Please clarify.

Line 42: Change "achieved" to "undergone".

Lines 44-45: Change to "Populations at high risk to GAS infections include ...".

Line 49: Omit "across the globe".

Lines 54-55: Change to "Antibiotics effectively control and treat ...".

Lines 57-58: Omit "Hence, antibiotic resistance is an inevitable challenge in the reduction of antibiotic sensitivity against GAS pathogens."

Line 61: Change "introduces" to "highlights".

Lines 74-76: Change to "Rheumatic heart disease is a commonly acquired heart disease caused by untreated or recurring GAS infection, and is associated with high risks of morbidity, disability, and mortality."

Line 84: Change "Inlander" to "Islander".

Lines 122: Change "antiphagocytic functions" to "phagocytic functions" if applicable.

Line 133: Change to "charged".

Line 147: Change "expressed" to "had".

Line 150: Change to "Immune reaction to GAS M protein provides protection against GAS infection, which encourages ...".

Line 153: Omit "which are responsible for antigenic variation".

Line 157-158: Change to "... N-acetylglucosamine, making the mucoid colony morphology smooth, watery, and of regular contour."

Line 170: Change "with the consequent induction of" to "resulting in".

Lines 180-183: Change to "Although the hyaluronic acid capsule is considered a weak immunogen since it induces anti-hyaluronate antibodies, it is important to note that anticapsule antibodies do not have opsonic properties. This means that they are unable to protect against infections by neutralising the antiphagocytic effects of the capsule."

Line 191: Change "secret" to "secrete".

Line 202: Change " SL) to "SLO".

Lines 211-212: What do you mean by ", achieving the apoptosis of host cells while no effect on the SLO-mediated portion of synthetic cholesterol pore"?

Lines 238-239: Change to "As a super-active protease with endopeptidase activity, SpeB contributes to the degradation of host extracellular matrix, ..."

Lines 267-269: Change to "Despite GAS vaccine research since the early 20th century, no commercial GAS vaccine has been developed."

Line 281-285: Change to "M proteins, the predominant virulence factor of GAS infections, are the most common pathogenic antigens of GAS vaccine candidates in both preclinical and clinical phases. Other vaccine candidates that are in preclinical trials contain numerous non-M protein virulence factors such as various enzymes, erythrogenic toxins, and capsule acid."

Lines 354-355: Change to "The hypervariability of the N-terminal region poses a challenge in the development of a vaccine against all serotypes of GAS."

Lines 360-361: Change "Since p145 was not the minimal sequence for GAS peptide-based subunit vaccines, it still induced autoimmune responses" to "However, it induced autoimmune responses."

Lines 451-451: Change to "Group A Streptococcus infections have a high prevalence worldwide."

454-455: Change to "However, since 2021 some high-income countries have also experienced sudden outbreaks ...".

457-460: Change to "Advances in GAS vaccine development in preclinical

and clinical settings have shown in the last decade that M protein-based GAS vaccines have a greater potential than non-M protein-based vaccines."

Comments on the Quality of English Language

 I have made some suggestions for changes in the text.

Reviewer 2 Report

Comments and Suggestions for Authors

Just a suggestion, Figure 1 & 2 can be merged which might be more efficient to present the strucutes. 

Lines 349-350, different font.

Reviewer 3 Report

Comments and Suggestions for Authors

This review briefly summarises the burden of GAS infections, followed by a short description of select virulence factors. The review then provides a comprehensive summary of the most recent vaccine candidates in preclinical development and a shorter description of those in clinical trials.

Overall, I was concerned with the need for more accuracy in several statements, particularly in the earlier sections. The review would also benefit from some thorough proofreading for grammar, which I found at times led to these inaccuracies. Inappropriate referencing, e.g. referencing of reviews for specific research, was also seen.

To add focus/relevance to the virulence factor section, I suggest concentrating on those factors that have been described in your vaccine candidates section.

Although I found the review lacking some novelty (there are many GAS vaccine reviews published recently), I thought the overview of the more recent preclinical vaccine candidates could be useful to the field.

Specific comments

Line43: I’m not sure what “metabolic type” means, also change “(MT1)” to “(M1T1)”

Line 52: Additional references are needed to make this generalisation.

Line:57 I would argue that antibiotics are useful for ARF, perhaps not for treatment per se, but at least for prevention of recurrent episodes.

Line 68: necrotising fasciitis is mistakenly placed as non-invasive

Line 71-72: Incorrect stats – I believe this is for invasive diseases only not all GAS disease.

Line 136: this sentence needs revising. Which parts do you mean? You follow by saying C-term is highly conserved not variable, so it is confusing.

Line 140: Incorrect statement, refs?

Line 146: Incorrect statement, GAS disease is not caused by M-proteins. These M serotypes are associated with disease

Line 149+: Table 1, I think you mean : “GAS M-protein serotypes” or just “M serotypes”. Some inconsistent referencing in this table e.g. ref 38 seems to be nothing to do with RHD. The paragraph following the table has some bold statements without adequate referencing to back them up.

Line 159: switch it around: the genes are involved in the production of the capsule

Line 185: why “non-protein”? All these toxins are proteins.

Line 184: section title needs revising as not all the toxins described in this section are pyrogenic.

Line 202: SL) = SLO

Line 203: “involved in the pathogenesis of GAS expression” – not sure what this means

Line 215: I wouldn’t consider a superantigen “a typical immunostimulatory molecule”. These toxins also importantly bind to the TCR.

Line 223: more suitable heading required.

Table 2: I suggest also adding the text described enzymes to the table as well for an overview.

Line 273: “various microorganisms” incorrect

Sentence starting 283 to revise as it doesn’t make sense.

Table 3 sections are difficult to visualise. Perhaps some shading of each section heading might help?

Comments on the Quality of English Language

See above

Round 2

Reviewer 3 Report

Comments and Suggestions for Authors

The authors have addressed most of the reviewer's comments adequately, although some minor (mostly grammatical) concerns still exist:

Line 41-42 is still confusing. It reads like GAS invades the host immune response in one host causes it to disseminate into a 2nd host. I don’t believe this is what you are trying to portray.

Line 48: ”S. pyogenes” should be italicised. “is survival” change to “can survive”

Line 64: Remove “however”.

Line168: The idea of rheumatogenic strains are quite controversial and geographically distinct. Many newer studies in areas where ARF is high show very large strain diversity, more reflective simply of what is circulating in the community at the time.

Line 170: “The essential virulence of GAS..” should read “The essential virulence factor of GAS..”

Line 181: I still don’t think the title is reflective of the content of this section. E.g.  Hyaluronic acid capsule is described in detail in the first part of this section and is not a pyrogenic toxin.

Line 206: “it induces anti-hyaluronate antibodies” ??? HA is produced in vast quantities by the human host and most of us will be tolerant to HA and not produce an immune response. This is how the capsule is used by the bacteria to hide itself from the human immune system.

Line 238: I don’t understand the new addition. Why would you expect NAD-glycohydrolase to enhance synthetic the SLO-mediated portion of cholesterol-rich liposomes? Do you mean the lytic activity mediated by SLO?

Line 252: suggested to edit to “Enzymes involved in GAS pathogenesis” as this section does not as it stand describe any studies describing these as vaccine targets.

Table 2: Suggest to add the enzymes in your text to the table. i.e. add streptokinase, Sep B, C5a peptidase, to your table as well.

Comments on the Quality of English Language

See above
